# Collaborative Action for Community Resilience to Climate Risks: Opportunities and Barriers

**Olivia Jensen** [1,2,*] and **Corinne Ong** [2]

1   LRF Institute for the Public Understanding of Risk, National University of Singapore, Innovation 4.0,
    3 Research Link, Singapore 117602, Singapore
2   Institute of Water Policy, Lee Kuan Yew School of Public Policy, National University of Singapore,
    469C Bukit Timah Road, Singapore 117602, Singapore; sppcopp@nus.edu.sg
*   Correspondence: olivia.jensen@nus.edu.sg

**Abstract:** Collaborative actions between citizens and government agencies at the local level have the potential to raise community resilience to climate risks via social learning and increased social capital. However, for these actions to succeed, community members' values, norms and risk perceptions must support collaborative action and be accompanied by self-efficacy. This paper develops a theoretical framework linking these concepts and applies the framework to a collaborative climate risk project in Singapore, using qualitative data from focus group discussions. We find that pro-social values are a strong enabling factor for residents to engage in collaboration, but that perceptions of low collective efficacy are a potential barrier. In particular, we find that the relative competence of the government leads to a perception of "exemptionalism," which undermines individuals' intentions to devote resources to collaborative action.

**Keywords:** community resilience; climate risk; social learning; social capital

---

## 1. Introduction

Global climate change is expected to generate and exacerbate risks such as flash flooding, heat waves and storm surges at the local level. In order to build communities' resilience to these risks, tailored small-scale interventions will be needed, in conjunction with efforts to adapt to and mitigate climate risks at the national and global scales. In this context, we understand "community" as a spatially bounded phenomenon or experience, comprising members of a local population sharing a common sense of identity, while striving to satisfy its daily needs and resolve shared problems—a definition drawn from community field theory [1,2]. Community resilience, meanwhile, we understand as the community's capacity to withstand and recover and learn from climate risk events, quickly and inclusively [3].

Local collaborative actions that bring together residents, government agencies and other stakeholders to design and implement interventions hold potential to raise community resilience through several channels. Firstly, collaborative actions can improve the public's understanding of climate-related risks so that they can make better decisions as individuals. Secondly, these collaborations can facilitate the development of shared understandings of complex socio-ecological problems. These shared understandings, which may be characterised as a form of social learning, are especially valuable under conditions of uncertainty like those surrounding climate change drivers, impacts and solutions [4]. Thirdly, collaborations open up opportunities for residents to create bonds with one another and with government agencies, contributing to social capital, which helps individuals and groups to resist shocks and recover more quickly and completely from disasters [5]. These attributes

of collaborative action to address local climate risks are recognised by stakeholders, and a growing number of government agencies across geographies are exploring projects of this kind.

However, while collaborative approaches have great potential, there may also be considerable barriers to their deployment. These may be due to perceptions of climate-related risks, values or norms that are inconsistent with collaborative work, or constraints on individual and community capacities and resources in the context of competing objectives.

This paper examines the scope for local collaborative interventions by exploring the enablers and barriers to their development in the context of a residential community, and comparing these findings with theories of community engagement. We expect our study to validate or enhance existing theories of community development and resilience, and to inform future climate initiatives that promote collaborative actions within communities.

An ongoing initiative to employ a collaborative approach to climate risk reduction in Singapore provides the context for this research. The initiative is described in detail in Section 3. Specifically, we conduct a qualitative analysis of perceptions, attitudes and intentions around climate risk and response in the locality of the collaborative intervention to better understand the *drivers and mechanisms* by which individuals' values, beliefs and norms are linked to community engagement on climate change (collective and individual) and the *opportunities and barriers* to enhance social learning and build social capital through collaborative actions.

## 2. Theoretical Framework

In this section, we set out a framework linking individual attitudes and intentions, via collaborative actions, to community-level changes in social learning and social capital.

### 2.1. Individual Attitudes and Self-Efficacy

Individuals' attitudes and intentions to engage in collaborative environmental actions and other pro-environmental behaviours are influenced by their values, beliefs and norms (VBNs) [6–8]. In relation to environmental behaviours, relevant values include environmental concern [9] and a sense of responsibility for environmental outcomes [10], while relevant beliefs include problem awareness and attribution relating to global and local climate change risks [11]. Related social norms could include types of public behaviour that are accepted and condoned by social reference groups, for example in relation to how community members engage in public sphere activities or norms relating to private sphere activities like consumption patterns or frugality [12].

Drawing on the theory of planned behaviour, self-efficacy or perceived behavioural control (PBC)—how easy or difficult an individual perceives an action to be to perform—is seen as mediating the influence of attitude on behaviour (Ajzen 1991).

### 2.2. Environmental Behaviours

Stern [13] provides a comprehensive classification of environmental behaviours that can be applied to climate risk resilience. He distinguishes public sphere behaviours, which are further classified into activist and nonactivist behaviours, private sphere behaviours and behaviours within organisations to which the individual belong. Nonactivist public sphere behaviours affect environmental outcomes both directly and indirectly. Direct impacts on resilience might include local collaborative actions to design a flood management strategy [14], or to adapt and adopt new agricultural practices and technologies [15], for example. Indirect impacts are made through influence on public policy. They include taking part in demonstrations, signing petitions, policy advocacy and support for pro-environmental policies.

Community engagement constitutes a group of actions and activities (e.g., projects, events, programmes) that may be ad-hoc or continuous and sustained by design, and are intended to involve local residents and mobilise action (Kaufman 1959). Drawing on community field theory, we expect that sustained and purposeful interaction in collaborative projects will enable social capital and social learning at the local level.

The community field extends beyond particular substantive, group-specific interests (also known as social fields), and targets the ability of an entire community to achieve a sense of shared identity and identification with the locality, trust, problem ownership, and translation of these into collective action and commitment to action [16], which is, in turn, expected to improve a community's resilience in problem solving, self-reliance in problem resolution, and identification with the locality (Lane et al. 2011).

### 2.3. Social Learning

Originally, social learning referred to learning by the individual occurring through social interaction rather than direct experience (Bandura 1977). Social learning's basic premise is that learning occurs through "vicarious, symbolic, and self-regulatory processes," where the mechanisms of social modelling and direct or vicarious reinforcement are significant. It also explains how "foresightful" or future-oriented behaviours are acquired by individuals through social learning processes (Bandura and Walters 1963; Bandura 1977). This is relevant to ecological behaviours whose positive or negative impacts may not be immediately observable by the individual.

In the socioecology literature, the term social learning has since been employed to refer to changes in understanding that go beyond the individual to the collective [17]. In this framework, social learning refers to widespread learning through social interaction that generates benefits for the broader socioecological system [18]. Here, we understand it to refer to learning by a social entity rather than an individual, and to a societal level search and learning process.

Social learning is thought to lead to better, more sustainable decisions and interventions than top-down decision-making when there is considerable variation in local conditions, both in the nature of the problems faced and in the range of potential solutions to them, in conditions of continuous change, when interventions need to be recalibrated and reviewed frequently, and when public support for and acceptance of interventions is necessary for their adoption and maintenance over time [19]. It is increasingly seen as necessary for the identification of sustainable and resilient solutions to complex and dynamic socio-environmental problems, as studies have exemplified (Adger 2000; Tidball et al. 2010).

The process of social learning begins with the collective definition of problems, their framing and the definition of boundaries, through identification and implementation of responses and the rules and practices that structure this process. The capacity for social learning depends on whether the group is able to deal with differences in perspective, solve conflicts between members, make and implement collective decisions and learn from experience. Social learning in turn raises resilience by increasing the capacity of the group to solve future problems.

### 2.4. Social Capital

The concept of social capital is a contested one that has undergone multiple definitional iterations and applications across different research contexts. Amongst these versions, we focus on Coleman [20] and Woolcock's [21] conceptualizations, and review empirical studies of social capital in the context of crisis.

Coleman [20] conceptualized social capital as encompassing obligations, expectations and norms; it is accompanied by sanctions inside and outside the family and influences individual level performance. On the other hand, Woolcock (2002) defined social capital according to three categories: bonding, bridging and linking.

Bonding social capital refers to strong emotional connections among individuals, in the family or among close friends [22]. Bonding social capital is often characterised by high levels of similarity between individuals in terms of their demographic characteristics. These strong bonds could provide support and assistance in times of need [23]. Bridging social capital describes looser horizontal ties between individuals and groups in clubs, associations and joint activities, including political, civic and religious organisations and interest or activity-focused clubs. These associations are likely to exhibit greater socioeconomic and demographic diversity and have been shown to be important in generating

economic opportunities [24]. Linking social capital refers to vertical ties between individuals and communities and formal institutions of government [25]. These connections could be important to getting access to services and resources both in the short-term context of emergency response and in the longer term for communities [26].

At the individual level, family members are often providers of critical information (warnings) and assistance (finding water, food, shelter and childcare) as first responders [23,26,27] to provide basic needs—food, water, shelter—and services, like childcare [23,28,29]. Those who are isolated and have few social ties are less likely to be rescued or receive assistance from others, such as shelter. Perhaps more surprisingly, they are also found to be less likely to seek medical help, or to take preventative actions such as evacuating [30]. In the recovery phase, bonds with friends and a feeling of belonging in the community lessens the mental and physical harm due to a disaster. To illustrate, households with larger Spring Festival networks were more likely to rebuild their home after the 2008 Wenchuan earthquake [31].

At the community level, empirical studies have found that measures of social capital including number of and participation in community organisations, length of residence and voter registration and participation rates have strong explanatory power in relation to the speed and completeness of post-disaster recovery [32].

Despite the prevalence of studies illustrating the benefits of social capital for individuals, communities or nations, Portes [33] cautions against an uncritical romanticizing of social capital. This, he argues, could lead to a biased and unscientific dismissal of social capital's potentially perverse effects. Following Portes, other studies have gone on to report the downsides of social capital. These include, firstly, the risks of engendering exclusionary membership conducing to parties of collusion [34], and, secondly, the possibility of members exerting excessive control over other members' self-accrued privileges, or free-riding on other successful members for personal gains [33]. Thirdly, high-solidarity groups may be internally cohesive but also highly vigilant in supervising members' actions, which compromises members' freedoms and privacy. Lastly, highly cohesive groups that experience lower social statuses in society may seek to protect their subcultural identities in defiance of the mainstream, ostracizing members who deviate from their subordinated identities and thereby reinforcing a negative "contagion effect" [34,35].

Drawing together these elements, we develop a framework linking individual attitudes and behaviours to collective outcomes. The framework is presented in Figure 1. Individual VBNs influence attitudes, which, mediated by self-efficacy and social learning, drive environmental behaviours in the public and private spheres. Public sphere behaviours contribute to the development of social capital, which increases community resilience and feeds back to an increased understanding of environmental risks, self-efficacy and social learning.

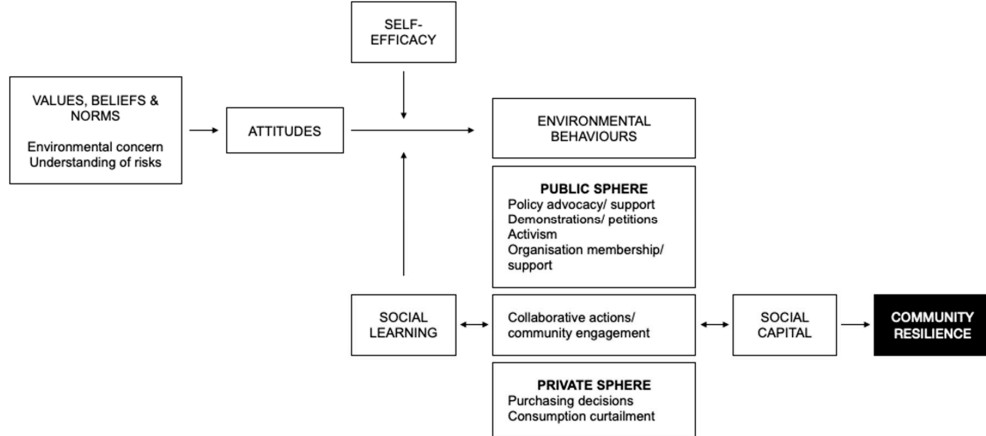

**Figure 1.** Theoretical framework: community participation and community resilience (Authors' elaboration).

Employing this framework, we can identify several possible barriers to collaborative action leading to increased community resilience:

1.　Underlying values may be inconsistent with pro-environmental behaviours: community members may set low value on environmental conditions and have a low level of environmental concern.
2.　Low perceptions of climate change risk [36]. This may be due to the amount or type of information that people have about climate risks (Lorenzoni 2007), the time horizon over which they assess risk or the heuristics, and biases that affect their risk perceptions (Portes 1998).
3.　Dominant social norms may be inconsistent with public pro-environmental behaviours.
4.　Individuals with a positive attitude towards pro-environmental actions may nevertheless not engage in those actions because they face time or financial constraints and competing priorities.
5.　The public or the government are unable or unwilling to engage in social learning with each other.
6.　If too few individuals across the community engage in the collaborative initiative, they will not build social capital either as individuals or at the community level.

Conversely, a community in which there is a high level of environmental concern, high perceptions of climate risk, supportive social norms, sufficient time and financial resources for individuals to engage, and openness to collaboration on the part of the government and residents is likely to see a positive effect on social capital and social learning.

## 3. Empirical Setting

Climate change is recognized by policy-makers in Singapore as an existential risk for the country, and government agencies are engaged in assessing risks and developing policies to manage them (GoS 2019). Singapore experienced a mean temperature increase of 0.9 °C in 2019 compared to the 1981-2010 long-term average—equivalent to the last warmest record of 28.4 °C in 2016. In 2014, the country also encountered a 27-day dry spell and, two years on, dry spells depleted a major reservoir and dams to an unprecedented 20% level. Flash floods, which caused major property damage, have also occurred since 2010 and are expected to increase in frequency in the coming years. Of particular concern for Singapore is the sea level rise. As a low-lying coastal island, Singapore could see its sea levels increase by 1 m or more by the year 2100 (Ministry of Environment and Water Resources 2020).

Concurrently, the government views climate risk as an area in which citizens should take an active role in identifying and implementing solutions. The Centre for Liveable Cities (CLC), a think-tank under the Ministry of National Development of Singapore, launched an initiative in October 2019 to promote active citizen participation in community-based climate projects, to be developed in collaboration with CLC and other government agencies. According to CLC, "It aims to foster greater ownership and resilience through active citizen partnership with the government and empower citizens to be involved in co-creating solutions to build up their adaptive capacity" (CLC, private communication). This collaborative intervention is conceived as a pilot project with the potential for replication.

The intervention is implemented over twelve months and involves an initial consultation with community members to generate project ideas such as small-scale green or blue infrastructures like raingardens, tree-planting or the greening of communal paved areas; the selection of low-cost, fast and tactical demonstration projects together with community champions, supported by CLC and other government agencies; and planning with community members on future project development and scaling up. The expectation is that, at the end of the intervention, community members will have the incentives and resources to self-mobilise to identify other community actions and to deliver and maintain these projects, including identifying additional financial resources necessary to sustain them.

The intervention is innovative in the Singaporean context as Singapore does not have a long tradition of public participation in policy-making. Singapore first moved towards consultative governance in the 1990s—a shift aimed at channelling advocacy initiatives and institutionalizing feedback channels [37]. Over time, public participation has deepened, as reflected in a rhetorical shift towards "co-creation" and "co-production" of public services [38]. This more collaborative approach

by government agencies is demonstrated in the involvement of civil society groups in the design of the Green Corridor, a former railway line developed as an urban linear park [39].

The site chosen for the climate risk collaborative initiative is an area of central Singapore called Pek Kio. This is a mature community with a mix of housing types, including public housing (buildings owned and maintained by the Housing Development Board), private condominiums and private landed properties, and residents have a range of socioeconomic characteristics and a population of approximately 6800. Residents are primarily of Chinese ethnicity, with small numbers of Malay, Tamil and other ethnicities. The area is primarily residential but also incorporates a fresh food market (a "wet market") and a food court with many small independent stalls selling cooked foods (a "hawker centre") and some small commercial premises. The area also contains public recreational facilities, including a children's playground, outdoor exercise equipment, a basketball court and a large open space marked for development and not accessible to the public.

Pek Kio is an area with comparatively high flood risk—it was this characteristic that motivated its selection for the first collaborative climate risk project in Singapore. In 2006-2015, parts of the area flooded 10 times. Since 2015, drainage works have reduced the flood incidence. These drainage works included enlarging and covering roadside drains. Singapore's national water agency, the Public Utilities Board (PUB), which is responsible for flood risk management, expects that more intense rainfall events in the future due to climate change will once again raise the flood risk in the area. However, there is little capacity to increase drain size further due to the density of development in the area. Other climate change-related risks include high heat levels and heat island effects, exacerbated by building density and limited greenery on main thoroughfares.

## 4. Methods

In order to understand the range of attitudes, beliefs and intentions of community members and their links to individuals' willingness to engage in collaborative climate risk interventions, we employed a focus group discussion method that was guided by grounded theory approaches to data collection and analysis.

Grounded theory is a qualitative method commonly used to support theoretical generation; it can be used alone or in tandem with quantitative (deductive) methods of verification. It consists of an inductive analysis of a social phenomenon that is often insufficiently understood and arriving at associations and explanations for the phenomenon through data obtained from observations and in-depth interviews with target populations. The constant comparison approach, an analytical strategy used in grounded theory, is intended to support hypotheses formulation and theory generation on a given phenomenon [40]. At the same time, grounded theory requires that researchers refer to ongoing material and data related to the study, to validate or expand preliminary analyses. This approach thus supports the generation of theory of social phenomena, beyond particularistic contexts.

Being a qualitative-inductive method, grounded theory's key limitation is that it does not provide confirmatory evidence, which is typically achieved by testing hypotheses on a sufficiently large sample [41]. Furthermore, as grounded theory is usually applied in the study of highly contextualized settings, its cross-context generalizability warrants verification through further studies [42].

Nonetheless, multiple studies of community resilience to environmental risks have adopted grounded theory in studies of communities, particularly if no prior research exist on them and a deep understanding of these contexts is needed for informing interventions [43–47]. Weighing grounded theory's limitations against the purpose of this study, we consider this approach to be apt nonetheless, given our intent to explore and explain in-depth the conditions within a flood-prone community that contribute to members' resilience to environmental risks.

To reflect the importance of place in our conception of community, we extended the typical focus group discussion (FGD) approach. Discussions were undertaken in two stages, first during a guided walk through the neighbourhood led by trained facilitators from a non-profit organization specializing in community engagement, during which the research team acted as observers; and second in a seated

discussion facilitated by the authors. The walking route was discussed and agreed jointly by the facilitators and the research team, while the discussion protocol was developed by the researchers. Each stage lasted for approximately 1 h.

This approach combining the walk and FGDs was adopted to collect information not just to gather the stated attitudes, beliefs and intentions of community members, but also to directly observe the interactions between people and their environment and the dynamics of groups of community members, which provided additional information on social capital and social learning. The neighbourhood walk allowed the facilitators to refer easily to features of the local environment and related activities, and for the authors to observe residents' interactions with physical spaces as well as intergroup social interactions.

Three FGDs were held with a total of 15 residents of the locality. Two discussions were held in English and one in Mandarin. They took place in October 2019. Participants were recruited via a convenience and purposive sampling strategy. Local volunteer groups, known as "grassroots organisations," were contacted and asked to recruit participants of diverse socioeconomic profiles within the spatial boundary of the community. Participants were drawn from among the grassroots membership and from the wider community.

Participants were aged between 36 and 64 years old, with an average age of 54; 60% were women. Participants represented a range of socioeconomic characteristics and housing types: 9 lived in public housing (HDB), two lived in private condominiums and four in private landed properties. All participants in the FGDs were property owners.

The FGD discussion protocol covered risk perceptions, environmental behaviours and members' expressions of a sense of community (emotional connection, identification); their volunteerism and extent/type of participation in the local community to resolve locally based, shared problems, as well as their attribution of responsibility in addressing climate-specific issues of adaptation/mitigation. The protocol is provided in Appendix A. As a semistructured FGD, the protocol provided scope for participants to raise additional topics beyond the protocol, as well as latitude for respondents to elaborate on topics covered in the protocol. Discussions were recorded, transcribed, and translated into English where necessary before analysis. Grounded theory analysis was the method employed to code and systematize the codes, from which we developed key themes to categorize, as well as relationships between categories (Glaser and Strauss 1968).

## 5. Findings

### 5.1. Values

*Pro-environmental values* such as love for nature and the desire to live in a "green" neighbourhood were clearly expressed by participants. Residents expressed their appreciation even for small greening initiatives such as a "Green Linkway" (a covered walkway between the bus shelter and housing blocks alongside which a trellis has been planted with climbing plants). During the walkabout, one participant commented:

*"I love everything about this place. I like to look at the flowers and greenery around here"*

Government interventions to address other urban management objectives were not seen as taking adequate account of unintended negative environmental impacts. Residents also expressed sadness and resignation about the removal of trees on one of the area's main thoroughfares during a project undertaken by the national water agency to upgrade the area's drainage. They also bemoaned what they perceived to be the overly radical pruning of other trees in the neighbourhood that had previously attracted a wide range of birds, which in turn had drawn birdwatchers and photographers among the residents.

*"Talking about trees: I don't know why [government agencies] cut so many, especially when the leaves grow out it is good because it produces oxygen and we need that. But I don't know why NEA [National Environment Agency] cut down so many. I think it's island-wide."*

Turning to *pro-social values,* these were expressed strongly by participants. They made frequent reference to benevolence, empathy and a desire to be connected to others:

*"There are a lot of seniors on the block, so we do try to look out and help them . . . . We always ask 'how are you doing?' to the point we even know what disease they have, so we can keep a look out for them."*

*"I realised that I spent time doing works to serve others, and it made me happy rather than staying at home doing nothing."*

These prosocial values had led some of them to volunteer in the grassroots organisations, and motivated others to check on and help their neighbours as an individual initiative, outside an organisational structure.

*5.2. Beliefs: Climate Risk Perceptions*

Most people in the FGDs agreed that climate change was a concern and would affect Singapore, although the timeframe was not clearly specified in their remarks.

*"It is unavoidable. The ice is melting and all the seawater will come rushing in, so it will definitely affect us."*

No participants expressed scepticism about climate change science or about the anthropogenic causes of climate change.

Participants identified a range of risks associated with climate change: sea level rise and associated risks of flooding in some coastal areas; higher temperatures; increased breeding of mosquitoes, associated with a higher prevalence of the mosquito-borne disease dengue fever; and the possibility of more frequent or prolonged droughts. The main sources of information were government communications and the mainstream media. Several referred to the Prime Minister's National Day rally speech in August 2019.

Striking in its absence was any reference to heavier rainfall events and the resulting flood risks. This may be due to the significant investment made in the locality over the last decade to widen and cover drains and to raise the road, which has reduced the incidence of flooding dramatically. It was clear that these community members had no knowledge of experts' forecasts of higher flood risks in this area in the future.

Heat was identified as a top environmental concern during the walks and in the sit-down discussions. Community members perceived that temperatures had risen in recent decades and remembered, in particular, cooler evenings and nights. They remarked on the physical discomfort caused by heat to themselves and to elderly residents in the community. Heat was perceived as a current and ongoing environmental problem in the locality and was not necessarily connected with climate change. This is consistent with expert views that higher night time temperatures in Singapore are driven by an urban heat island effect rather than by global climate trends [48]. However, participants considered their own understanding of climate risks in Singapore to be limited.

*"I think we need to be educated, but we are not sure what will happen, so we need more information to be shared. Like a bit more transparent, like what exactly are we looking at. But also, we are also not sure what is the situation on understanding the climate change, you know like whoever is in charge researching and all – there is no plan, like how bad is the situation, or how many years are we left with it, there's all these question marks. So yeah, I think as citizens we don't know where to go, we do our best . . . "*

Having more "transparent" information on the level of risk that one should be concerned about, in other words, risk communication, as well as how exactly to respond to the risk, were considered important to residents if they were to be mobilized for climate adaptation or mitigation.

### 5.3. Social Norms

The discussions did not reveal any tensions between prevailing social norms and participation in collaborative community projects among the participants. Women and men, across races and ages, were comfortable taking part in public, group and outdoor activities in the community, including cultural and physical activities. Participants did not signal any dissonance between their perceptions of others' expectations of their behaviours and their efforts to engage in individual environmental behaviours. They also felt comfortable questioning some actions of government agencies (with regard to tree pruning, for example) and expressing their views on particular policies (such as taxes on electrical cars, which they saw as a barrier to the take-up of this technology in Singapore).

Participants were generally proud of their efforts to engage in individual pro-environmental behaviours and distinguished themselves from other community members who did not make similar efforts, for example in relation to recycling household waste, where some participants had sought to encourage other residents to use communal recycling facilities and noted with some frustration the low level of recycling in the area.

> "Communities are not easy to change. Because we are used to throwing our rubbish down, so convenient. I bring the plastic bag of bottle, inside [the recycling bin] wood, trash, whatever Styrofoam box ... They just throw everything inside."

### 5.4. Self-Efficacy

In the discussions, many participants signalled low perceptions of their ability as individuals to participate in collaborative projects due to their lack of knowledge and skills. Knowledge of both the problem and potential solutions were seen as residing with government agencies, and participants did not feel that they had relevant knowledge to bring to the process. Rather, the information flow was perceived as unidirectional, with the government as the information source on both the nature of the problem and potential solutions. When asked specifically about the roles that they could play in community-level projects, participants saw their role as limited to the provision of feedback on designs generated by others.

> "We can get involved, but we may not be able to do hands on. We can give you some idea, I can give you a small part and [government agencies] expand it from there."

Another echoed this view, suggesting that it was obvious that community members would not be able to take on a more far-reaching role:

> "We can contribute ideas, but design of course we can't do."

Even amongst those who were ready to lend support to the cause of climate adaptation or mitigation, specific directives and leadership guidance could be integral:

> "We are all on standby, we are all eager to contribute. As long as you give us a direction, what should we do, how should we do, I think we need that leadership. Because we have the community, we have the power and support, so what would you like us to do?"

Their perceptions of their own limitations were generally articulated alongside an evaluation of government agencies' high level of capacity; in other words, perceptions of the government's "hyper-effectiveness" seemed to produce a sense of lower self-efficacy and initiative among participants. This created a sense of "exemptionalism," in that participants perceived that they were exempt from

the need or responsibility to intervene and participate in problem-solving, as well as having little to no ability to intervene. The sense of exemptionalism was observed in at least two participants' quotes below:

> *"I think that the government will do something. In the future the government will have definitely prepared some things to help it."*

> *"Because we have a very good government, we are of the mindset that they will take care of it."*

One participant explicitly critiqued this attitude of exemptionalism displayed by citizens in assuming that all problems can be resolved by the government, and qualified that this represents insufficient effort by authorities to convey the risk and involve citizens in problem solving:

> *"Again in Singapore we are very fortunate, because we have a very good government, we always have the mindset that they will take care of it. But I think it is not enough, because the government cannot solve all these problems by themselves, so I think that level of communication is absolutely not there at all."*

Collective efficacy was also low. Participants were highly sceptical of their capacity, or the capacity of the country as a whole, to take actions that would make a difference.

> *"This is an international issue and we are a very small country. If China and America do not fall into line, if the two big countries don't do anything, then it's jialat liao (too bad)."*

Participants did not identify any role for Singapore as an advocate for climate action, despite the country's efforts to position itself as a leading example of green urbanism [49].

### 5.5. Pro-Environmental Behaviours

Participants offered numerous examples of pro-environmental behaviours taken at the individual and household level, mainly relating to energy and water efficiency:

> *"I try not to use aircon. My house is designed such that there is cross-ventilation so even when it is very hot outside my house is very cool. I don't like to use aircon; it is one of the things that if we can restrict the use, we can bring down the energy [use] and bring down the emissions."*

> *"I started telling my kids, stop taking long showers! I am going to time you!"*

Although none of the participants had themselves made a major household expenditure decision based on environmental concerns, such as deciding not to purchase a car, or to buy a hybrid or electric car, they noted that there was a generally low level of car use in the area. They remarked on the convenience of public transport, noting the exception of those with young children. This also prompted participants to see little value in trying to reduce car use in the area further, for example by instituting a no-car day. One participant expressed enthusiasm for the adoption of electric cars, but saw this in terms of policy advocacy rather than as an individual purchasing decision that they intended to make.

Participants did not report engaging in collective pro-environmental actions such as signing petitions, joining demonstrations or lobbying activities, engaging in environmental activism or belonging to or donating to environmental groups. With the exception of support for electric cars articulated by one participant, they did not mention specific environmental policies that they supported or disagreed with.

### 5.6. Social Capital

We found high pre-existing levels of social capital among all participants. Social capital was indicated by qualities such as participants' perceived closeness to neighbours, sense of trust in their

exchanges with neighbours, types and quality of exchanges with neighbours, and extent of familiarity with one's neighbours.

On average, the participants knew and interacted with five of their neighbours and said that they felt "close" to them. "Close" was defined as feeling comfortable borrowing household items or helping with household emergencies (power, plumbing). A number of the participants emerged as key community members, with a much larger social network. Notably, these key residents knew which households in their immediate area had elderly or infirmed members. One said that she knew 80% of the residents in her apartment block.

Being able to maintain harmonious relationships with their neighbours (being mutually friendly), maintaining camaraderie, and having someone close by to look out for them were the types of relationships participants desired achieving with community members. These were considered important as neighbours shared a common social-residential space and were likely to meet one another frequently, thus creating the impetus for relationship maintenance and mutuality.

In everyday matters, participants indicated a high level of trust towards their neighbours. For example, residents informed their neighbours when they went on a trip or entrusted items of value, such as house keys to neighbours when leaving for a trip.

The majority also participated in community events like Chinese New Year celebrations, and many were also involved in publicizing these events to other residents. Shared activities with neighbours included chatting, meeting for refreshments and taking exercise. One participant also ran daily exercise classes (tai chi) open to community members. Meeting neighbours incidentally as a result of one's children's activities was also common for members who were young parents. Among younger participants, especially those living in private apartments, communication with neighbours tended to take place online via chatgroups. On the other hand, older participants (middle-aged and older) tended to meet with one another in person, particularly in public areas within the community (e.g., the marketplace, exercise facilities around the estate, interest groups organized within the community).

There were also some gendered associations in terms of the activities that participants tended to do *with* and *for* their neighbours. The topics of discussion for females tended to be centred on the sharing of food within the community (e.g., "magic kitchen"), mutual help in childcaring responsibilities, and "looking out for" senior neighbours. On the other hand, male participants tended to engage in interest-based activities such as karaoke or exercise, and to offer mutual help in areas such as technical repair work, the sharing of mechanical tools, gardening, etc.

However, there were also limits, or, in fact, unspoken but acknowledged norms, on the types of conversations that could be comfortably had between neighbours even if they had high social capital. Participants did not report seeking help from neighbours on financial or health issues as these are culturally sensitive. When probed, participants shared that such issues were more private in nature, and participants would only disclose such matters to, or seek help from, family or friends.

> *"My previous neighbour, I was friends with her niece since I was young. So, if it really came down to it, if one day I really have no choice and I ask her, she will be willing to lend me money. But I feel that asking money from neighbours is really very . . . unbearable, there is a strong stigma against it, unless it is really dire or desperate."*

One older male participant attributed this to a cultural norm, whereby disclosing such personal matters in order to seek help would lead to one losing "face," suggesting that it was embarrassing for others to know such information about him:

> *"We Chinese are very concerned about our image. I think when it comes to financial matters, we will be very paiseh (embarrassed) to ask, but when it comes to situations like our neighbours needing an ambulance, or when there is a blackout, we will be willing to lend some lights."*

Participants explained that they had a good relationship with neighbours for multiple reasons. For some, their extraverted personality made them drawn to meeting and interacting with others,

which also led to their interest in participating as active volunteers in local community groups. For others, a relatively long history (>10 years) of residence in the estate increased their familiarity, trust, and opportunity for bonding with neighbours who were also long-time dwellers of the estate. The length of residence thus created a form of "social history" that could be shared between neighbours and facilitated bonding between them.

None of the participants lived alone, and key members explained that no one in their immediate area (block) lived alone (without another adult) either. This suggests that a very high proportion of residents in the locality have at least a basic level of bonding social capital.

Key community group members played a further role in connecting residents to government agencies.

*"I am close with [the block residents], so anything they feed back to me I will tell them not to worry and help to get it resolved."*

## 5.7. Community Volunteerism

Outreach to local community members for mobilization requires that barriers to volunteerism are recognized and taken into account. As identified earlier, low perceived efficacy constitutes one potential barrier to local participation.

Firstly, a lack of free time was found to be an important inhibitor to volunteerism in the community—a fact that is widely recognized in the literature on volunteerism. This is associated, in particular, with those who have pre-existing commitments to other activities beyond the community, or those currently assuming caregiving roles (e.g., the parents of young children). The culture of longer working hours in Singapore also poses greater difficulties for working adults who want to prioritize time for volunteerism amid other activities that demand for their time:

*"I feel that, in my case, if there is no interest I won't do it. And time constraint is an issue, if there is no time I won't do it. So time and interest is an issue. If there is no time to sleep, to spend with the family, how can they participate?"*

*"I think there are also some barriers, because people are generally working longer hours, and now with online social media and all that we can do a lot of things online, some people just use online as entertainment. And then of course there are demands from young children, and some people are trying to be healthy, they can't even find time to go to the gym, so there's just a lot of competition for time."*

While community volunteers could be "reimbursed" for their time and labour, which some participants suggested was important, others were hesitant to provide economic incentivization, which, they felt, defeated the purpose of volunteerism which ought to come "from the heart":

*"Money is additional, the best is if it comes from the heart. These kind of money things, if you give a lot it is bad, but if you give little people will not be willing."*

Other barriers to volunteer recruitment and retention may include physical and organizational barriers. In terms of physical barriers, one participant who has experience recruiting grassroots volunteers shared that private residential apartments that are gated, restrict entry from the public. This creates a physical challenge that limits door-to-door recruitment of community volunteers:

*"But I also notice right, we also have other problems reaching another group of people here. Landed is very easy, we can just go to their door and invite them. But condominiums, there are two types, one type where they have security guards, and then we cannot go in without permission. So usually we will try to recruit residents who are staying there, so for example, if I stay in the condo I can be the person who opens up access to the residents in my estate. But there are a lot of estates that we can't go in, those are private estates, we can't go in when we don't have members and there are security guards, it's very automated you have to press button – then you can't reach that segment. There are a few condos here that are like that, so we don't have access."*

On the organizational front, electoral boundaries, which are redrawn prior to each election, could lead to a dearth of volunteers that may impinge upon the very survival of one's membership:

*"The last time we had 25 to 30 committee members, but because of the redrawing of the boundary, one third of them either went to Toa-Payoh-Bishan or Jalan Besar [other localities in Singapore]. So now we have our fingers crossed for the next redrawing; [we don't know] whether we [will] disappear or not."*

Another participant added:

*"We used to be a much bigger group, and then our constituency got cut into three, so suddenly our strength dropped to about 1/3. So now we have only 12 regular committee members."*

The stability of an organization can be akin to an existential crisis for members, potentially subverting "locality group identity." This could have the effect of hampering volunteer recruitment and decreasing existing members' morale [1]:15), and thus qualifies as an important consideration when forming local informal institutions.

## 6. Discussion

The FGDs revealed the presence of enabling factors favouring collaborative actions to manage local climate risks. First, prosocial motivations are strong among the individuals who took part and are linked to behaviours like participation in community meetings and organization of events, which are relevant in the context of climate risk adaptation work and are similarly demanding in terms of time. This suggests that these values and the satisfaction derived from these volunteer activities is high enough to overcome competing uses of time and resources, at least among the sample of individuals in the FGD groups. Pro-environmental values were also in evidence, but were not as strongly connected with behaviour, and participants did not have any experience of motivating others to take part in environment-focused events and activities.

Potential barriers are also evident. Firstly, exemptionalism was a strong current in all of the group discussions and constitutes a significant barrier to collaborative projects. The competence of Singapore government agencies in reducing everyday risks like flooding has effectively reduced perceptions of these risks; people who have lived in the area for five years or less are unable to recall any instances of flooding in the neighbourhood. People naturally look to a government that has invested heavily and successfully in grey infrastructure over decades to offer similar solutions to climate change risks. This perceived competence generates a dilemma for the government, which wants to manage risk effectively in the short and medium term and to convey to the public the information that these risks are expected to rise in the future due to climate change and that the ability of traditional grey infrastructure solutions to deal with them may be inadequate or prohibitively costly.

Second, a low level of collective efficacy is found at the national-international level too. The study participants did not see a way for Singapore to effect change through international negotiations and so were not motivated to engage politically, for example by lobbying for carbon taxes or stricter regulations on emissions. This may be something that will change over time, perhaps as a result of media coverage or additional information.

Third, the potential for social learning through collaborative working will be limited by participants' lack of specialist knowledge of local climate impacts and appropriate responses. Even though participants signalled a willingness to engage, they did not have independent sources of knowledge about local problems or potential solutions. There seems to be a pervasive idea that the lived experience of floods, heat, etc. is less important to the planning of responses than the scientific or evidence-based assessment of risks conducted by government agencies. Rather, they relied on the government to provide them with information about these. Part of the problem may lie in a missing element of the intervention that is an essential part of social learning, the collective acknowledgement and definition of the problem; in this intervention, community engagement is limited to generating solutions to

predefined problems. Given the trust in local government institutions, there is a role for these institutions to mobilize communities to engage in community-led environmental causes—at least in the initial phases of an intervention. However, the fine balance lies in ensuring that communities have a stake in the problem and its solution, wherever possible, thus minimizing dependence on authorities to solve these problems.

The high level of social capital in the study community suggests that there is already considerable community resilience, in the sense of being able to withstand a risk event and recover quickly. Vulnerable members of the community were known and channels to share information and resources already appear to be in place. The high level of social capital may also act as an enabling factor for the information-provision aspect of the project. As participants in the FGDs already knew and engaged with a circle of community residents outside their own family units through social activities, recreation and day-to-day interaction with neighbours, this would make it easier for them to engage with and potentially mobilise other residents to get involved in projects. However, the high social capital stock will also make it more challenging for a collaborative intervention to demonstrate a positive impact on social capital in the locality, especially if there is self-selection among the participants, with community members with the highest ex ante levels of bridging social capital choosing to take part in the intervention. Therefore, the constitution of members will be a key consideration for democratizing membership and ensuring that the interests of diverse groups are represented and considered. This may require revisiting and addressing barriers to volunteer participation, which can be physical, psychological and/or organizational in nature.

We propose a revised model of collaborative action to take account of these factors. The revised model explicitly includes social concern and locality group identities as relevant VBNs and includes collaborative actions as one form of environmental behaviour. It also highlights the multiple aspects of perceived efficacy that are relevant in an intervention of this kind. The revised model is presented in Figure 2.

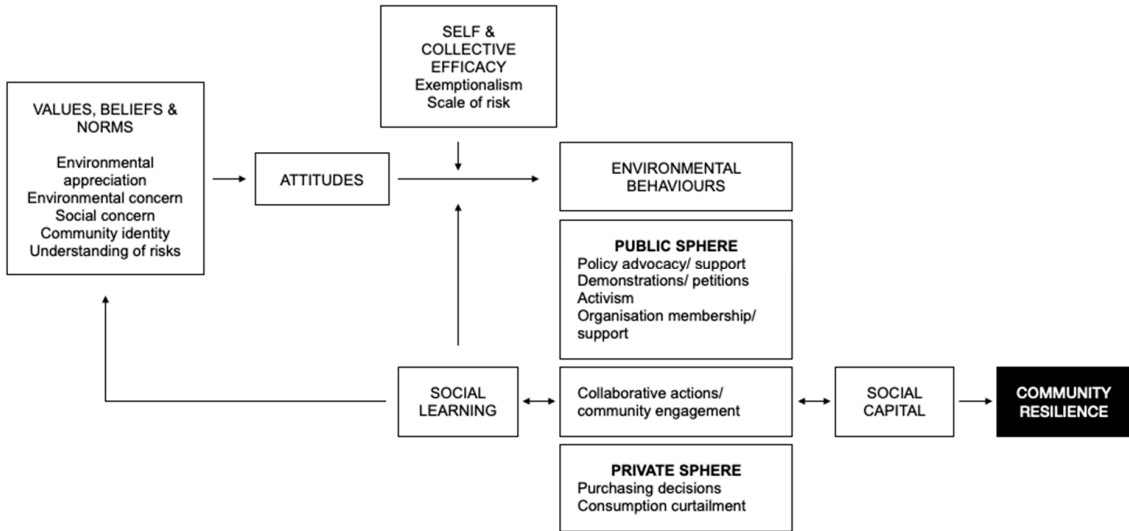

**Figure 2.** Revised theoretical framework: community participation and community resilience (Authors' elaboration).

The model can provide the basis for further exploration of VBNs on collaborative actions, as one form of environmental behaviour, in the design of survey instruments or experiments.

## 7. Conclusions and Recommendations

The analysis conducted in this paper is of course particularly pertinent to the design of community-level projects in Singapore, with its distinctive geographical, meteorological, social

and institutional characteristics, but some findings may be useful to stakeholders in other jurisdictions interested in the potential of collaborative actions to increase resilience to climate change.

The analysis revealed two important knowledge gaps among community members: local impacts of global climate change, and how this is expected to interact with other phenomena like urban heat islands and urban environmental management to shape risks at a very high spatial resolution; and knowledge of actions that individuals can take to manage and mitigate these risks. To a large extent, these knowledge gaps reflect a high level of uncertainty on these issues among experts, so they will not be resolved simply by increasing transparency. However, more can be done to design practical information and communications materials on these aspects of climate risk, acknowledging as needed areas of uncertainty in the science.

As pro-environmental values appear to be more weakly held than prosocial values in this community, an intervention may be more successful at mobilising participation if it is framed in terms of enhancing social benefits rather than environmental benefits. As our study has shown, it was in residents' interest to advance positive relationships with other neighbours in order to increase the capacity for mutual assistance and support where needed, and to maintain a harmonious relationship with their neighbours. These social benefits are likely to be inherent in a community with high social capital, and could be harnessed for other causes that residents would not otherwise, choose to devote time to. As illustrated in our findings, many young to middle-aged residents have competing priorities and limited time for volunteerism. Without a pre-existing interest in a cause such as environmentalism, they are unlikely to pursue volunteerism. As such, capitalizing on social benefits could be advantageous.

A further recommendation that echoes guidance for initiatives dealing with other forms of risk is to involve the community in problem definition at the outset. While government agencies, researchers and other experts may play a role in providing information on their perceptions of the problems the community faces, there may be a gap between these perceptions and those of the community themselves. If community members perceive risks associated with heat waves but not from flooding, for example, they are likely to be more willing to engage in collaborative action that addresses heat risks. With collaboration established, experts will have additional opportunities to share information about other risks that they perceive to be underestimated by community members.

Further research is needed to assess the value of the analytical framework proposed. The relationship between social values and pro-environmental behavioural intentions could be investigated through surveys, and the links between values and actual behaviours tested in an experimental format, ideally in future cross-country studies. We also recognize potential limitations in the application of findings from this study to other contexts. This is because the expression, or capacity for expression, of social capital is likely to be context-dependent. Cross-national studies have qualified that the development of social capital depends on multiple macro influences, including a country's political institutions, sociocultural norms [50]; extent of ethnic heterogeneity [51] and economic performance [52]. The results of our research in Singapore may be of particular relevance to other highly developed, ethnically diverse urban settings.

Further work is also needed to fully operationalise the concept of community resilience to climate risk to make it possible to evaluate the impact of collaborative actions on resilience across geographical, economic and sociocultural contexts.

**Author Contributions:** Conceptualization, O.J.; Methodology, O.J. and C.O.; Validation, O.J. and C.O.; Formal analysis, O.J. and C.O.; Investigation, O.J. and C.O.; Resources, O.J.; Data curation, O.J. and C.O.; Writing—original draft preparation, O.J. and C.O.; Writing—review and editing, O.J. and C.O.; Visualization, O.J.; Supervision, O.J. and C.O.; Project administration, O.J.; Funding acquisition, O.J. All authors have read and agreed to the published version of the manuscript.

**Funding:** This research was supported by the "Environmental Risk in Urban Asia" grant from the LRF Institute for the Public Understanding of Risk, grant number R-727-003-003-133.

**Acknowledgments:** The authors thank all participants of this study, and the community grassroots leaders who generously helped recruit participants for the study. We are also very grateful to CLC for the opportunity to conduct research on an ongoing initiative. We thank Claudene Tan for excellent research assistance.

**Conflicts of Interest:** The authors declare no conflict of interest.

**Appendix A  Focus Group Discussion Protocol**

Today we'd like to talk to you about your neighbours and your neighbourhood and the local environment. Thank you for taking part. Our discussion will take about one hour. Please let me know if you need to leave the discussion at any point.

To get started, let's get to know each other. Please could you introduce yourself to the person sitting next to you and tell them a few things about yourself. Tell them whether you like to take part in any recreational or leisure activities in this area around Cambridge Road and what those are, whether you do some of these activities with your neighbours and whether you are a member of any group or organization (outside your work) in this community or around Singapore. When you've both introduced yourselves, please could you turn back to the group and introduce the person you've just been getting to know to everyone else, using the information they have just shared with you.

Now we'd like to discuss some questions all together as a group. First, I would like to ask each of you:

1.  Do you know your neighbours' names? What about their telephone numbers? How did you come to know them?

2.  What is your relationship with your neighbours like? [Probe: Do you go to your neighbours for help, for example to borrow a household item, to receive a delivery or to leave a spare key with them? Do your neighbours sometimes ask you for help? Do you engage in recreational/leisure activities with them?]

3.  What does it mean to you be "close" to your neighbours? Can you give some examples?

4.  How about in a time of crisis? What do you think it would be like then? What kind of relationship would you like to build with residents in your neighbourhood? What would it take for residents in your community to achieve this kind of relationship? Do you think there are any barriers to achieving this? What do you think your role is in helping to build this kind of community?

5.  If you faced a big challenge, for example to do with your health or with finances, who would you go to for information? Who would you go to for help?

6.  Do you know your local grassroots leaders? How did you come to know them? How would you describe your relationship with local grassroots leaders in your community?

7.  How do you feel about your neighbourhood in terms of the local environment, space, facilities and amenities? What are your favourite spots in this neighbourhood? [Probe: What do you like about them? What kinds of activities do you like to do there? Do you meet family, neighbours or friends there?]

8.  Would you like the local environment/facilities/amenities to be different in any ways? What would make you want to spend more time in community spaces, rather than staying at home?

9.  Recently there was a lot of haze in Singapore. Did that affect how much you went out in the neighbourhood? How did it make you feel? Were you worried about it? Did it affect your health at all, or that of your family members? Did it have any other impacts on you and your family? Did it make you change your actions in other ways? How about for other members of the community, especially the elderly or people who are sick—do you think it affected them? Did you help them?

10.  Do you think the haze will return, and are you worried about that?

11.  Have you experienced heatwaves or floods as a resident here? How did they make you feel? Were you worried? Did they affect your health at all, or that of your family members? Did the extreme weather have any other impacts on you and your family? Did it make your change your actions in other ways? How about for other members of the community, especially the elderly

or people who are sick—do you think it affected them? What can you do to help them? What should others do to help?

12. Do you think there will be more heatwaves or floods in the future, and are you worried about that?

13. Have you ever experienced a drought in the time you've been residing here? What impacts did it have on the local environment and on you? Did you change your usual behaviours at all during the drought, e.g., how much you watered plants, how often you took showers, etc?

14. Do you think there will be more droughts in the future, and are you worried about that?

15. You may have seen in the news recently that sea level rise caused by climate change may impact Singapore. Did you see any news stories about that? How did they make you feel?

16. Do you think that climate change is going to affect Singapore? Would you like to know more about climate change in Singapore? What would you like to know?

17. Which sources of information about environmental issues and climate change do you trust, and why?

18. What do you think you as an individual or family can do about climate change? What would you be willing (and comfortable) to do, and what would you not be? Please describe your concerns.

19. How about communities—what actions are communities taking? What more could they do?

20. Do you know what the government is doing about climate change? Do you think that it is enough?

We are coming to the end of the discussion now. We've just got a couple more questions to talk about.

21. If you could imagine the government and communities coming together to prepare for future environmental issues, what could their cooperation be like?

22. Would you like to be involved in the design of an environmental project for this area? What would it take for you to be involved in any of these roles (go over each one)?

    a. A one-time meeting
    b. Regular meetings
    c. Community planning council
    d. Voting on options presented by experts
    e. Design or idea workshops
    f. Design competition
    g. Informed through regular newsletters
    h. Maintenance/oversight of infrastructure property/site

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
