# Peer review of "Collaborative Action for Community Resilience to Climate Risks: Opportunities and Barriers"

_sustainability, doi:10.3390/su12083413_

Round 1
Reviewer 1 Report
Dear Authors,
very compliments for your paper. Its results are very interesting. Generally, the manuscript has good writing but some sections need improvements.
If you take under consideration the remarks indicated in the notes below, I think that your manuscript can continue in the publication process in Sustainibility Journal.
31-33: Please provide some literature or explanation, why you define community in this way? I understand the community/society distinction in a different way than proposed here (see i.a. Ferdinand Tonnies' 1887 essay).
64: Why you didn't operationalize the definition of community resilience?
70: I can't agree with this. There is one dominant social learning theory by Bandura and Walters (1977), later developed. Please reformulate this section and add more literature review.
94-96: what is the purpose of this section? Will you analyse the role of leadership in social learning, I didn't find any appropriate paragraphs in your research and conclusions section?
99-113: I advise to introduce the critical approach to the social capital by Alejandro Portes (to better understand Putnam).
170: I'm not sure, this category is needed. Instead, I would rather expand the description and operationalization of social learning at the individual level.
181-182: I'm not sure, this figure is correct. I have two doubts: I think, that social learning should be at a similar position as self-efficacy. Second: The social capital (and social learning too) is not the outcome of mentioned behaviours of an individual. Please reconsider it.
283-286: if FGDs were held in the English language, what was the ethnicity of participants? How it fits the ethnic diversity of the analyzed neighbourhood?
464: I suggest "actions" instead of "behaviours"
600: Too many brackets
624: "collective self-efficacy" sounds bizarre to me. Collective efficacy or general self-efficacy sounds better, I think. Self-efficacy is a feature of an individual, cannot be collective, but can be aggregated to the societal level (generalized).
664-666 - the same remarks to figure 2 as to figure 1 (and the collective self-efficacy is also used in this figure, see comment above).
701 - because social capital seems to be an important factor, it is advised to notice, that the level of social capital is different in different national cultures. You can check for example the cross-national surveys about generalized trust (one of the most common indicators of social capital). What is possible in Singapore, could be difficult in Romania or Greece, because of the cultural differences.
Other general remarks:
You missed community resilience in "discussion" and "conclusions" sections. In my opinion, you should refer to it, somehow.
Reviewer 2 Report
The manuscript 'Collaborative action for community resilience to climate risks: Enabling factors and barriers' by Jensen and Ong tackles a relevant problem for securing the sustainable development of our society. However, the manuscript needs significant improvements before acceptance for publishing.
The study is very much a qualitative approach and is must justified much better the benefit of such studies.
It is not clear what is the rationale of this paper. Why do you consider it important? Are the results transferable to other areas? The methodology is unclear in this respect.
The use of the word 'factors' in the title is ambiguous. What factors do you refer to?
Is it climate change an issue in Singapore? Please elaborate more on this topic, provide some examples and statistics.
Please introduce some paragraphs to emphasize the limits of this method.
L56-57: what initiative is this about? could you provide more details?
The introductory part of the theoretical framework should be more elaborated. Are the sub-categories mentioned here the only direct and indirect ones? Please explain more detailed the difference between direct and indirect actions in relation to climate risk resilience. Provide some examples of good practice and citations.
L277-278: Please detail the method of sampling and quote similar approaches.
L358: this sentence is a corrolar of the study and it must be explained bot in a general manner and in particular. If the undersanding of the participants of climate risks and, probably, of climate change is limited, how this study can be relevant in any way?
L678: This key recommendation is very general and well known.
Round 2
Reviewer 2 Report
The authors have provided satisfactory feedback and I agree with the present for of the manuscript for being published in the journal Sustainability.